# Effects of Antioxidants on Pain Perception in Patients with Fibromyalgia—A Systematic Review

**DOI:** 10.3390/jcm11092462

**Published:** 2022-04-27

**Authors:** Ana Fernández-Araque, Zoraida Verde, Clara Torres-Ortega, Maria Sainz-Gil, Veronica Velasco-Gonzalez, Jerónimo Javier González-Bernal, Juan Mielgo-Ayuso

**Affiliations:** 1Research Group Pharmacogenetics, Cancer Genetics, Genetic Polymorphisms and Pharmacoepidemiology, Department of Nursing, Faculty of Health Sciences, University of Valladolid, Campus of Soria, 42003 Soria, Spain; zoraida.verde@uva.es (Z.V.); veronica.velasco.gonzalez@uva.es (V.V.-G.); 2Department of Nursing, Faculty of Health Sciences, University of Valladolid, 42005 Soria, Spain; clarajaen@yahoo.es; 3Emergency Service of the Hospital Santa Bárbara, Soria Healthcare Management, 42005 Soria, Spain; 4Recognized Research Group “Pharmacogenetics, Cancer Genetics, Genetic Polymorphisms and Pharmacoepidemiology”, Department of Cell Biology, Histology and Pharmacology, Faculty of Medicine, University of Valladolid, Center for Drug Safety Studies, 47005 Valladolid, Spain; maria.sainz@uva.es; 5Department of Health Sciences, Faculty of Health Sciences, University of Burgos, 09001 Burgos, Spain; jejavier@ubu.es (J.J.G.-B.); jfmielgo@ubu.es (J.M.-A.)

**Keywords:** antioxidants, fibromyalgia, pain, supplementation, systematic review

## Abstract

In recent years, antioxidant supplements have become popular to counteract the effects of oxidative stress in fibromyalgia and one of its most distressing symptoms, pain. The aim of this systematic review was to summarize the effects of antioxidant supplementation on pain levels perceived by patients diagnosed with fibromyalgia. The words used respected the medical search terms related to our objective including antioxidants, fibromyalgia, pain, and supplementation. Seventeen relevant articles were identified within Medline (PubMed), Scopus, Web of Science (WOS), the Cochrane Database of Systematic Review, and the Cochrane Central Register of Controlled Trials. This review found that antioxidant supplementation is efficient in reducing pain in nine of the studies reviewed. Studies with a duration of supplementation of at least 6 weeks showed a benefit on pain perception in 80% of the patients included in these studies. The benefits shown by vitamins and coenzyme Q10 are remarkable. Further research is needed to identify the effects of other types of antioxidants, such as extra virgin olive oil and turmeric. More homogeneous interventions in terms of antioxidant doses administered and duration would allow the effects on pain to be addressed more comprehensively.

## 1. Introduction

Fibromyalgia (FM) is a syndrome characterized by chronic widespread pain [1]. Pain is the predominant symptom; allodynia and hyperalgesia are also frequent [2]. These patients also present severe fatigue, impaired cognition, and sleep disturbance, among others [3,4].

In the 1990s, the American College of Rheumatology (ACR) approved criteria for diagnosing fibromyalgia. These criteria, in addition to chronic pain, established eighteen other areas of tenderness including chronic widespread and skeletal pain. Their duration had to be longer than 3 months and they would be positive if during the examination the tender points were positive with pressures of 4 kg/cm^2^ [5]. It was then that this disease shifted from a musculoskeletal to a neurobiological focus [6,7].

The diagnostic criteria for this pathology established in 2010 focused on the widespread pain index (WPI) and a symptom severity score scale [SS-Score] [8]. According to published evidence, this method allowed 88.1% of cases diagnosed by the 1990 ACR criteria to be correctly classified. This makes it easier and more comprehensive to determine the diagnosis based on patient information [7].

Unfortunately, conventional medical therapies targeting this pathology produce limited benefits. Review studies suggest that the combination of pharmacological and alternative therapies (including heat and light treatments, the use of bioactive plant molecules, electro stimulators, and body exercises) can improve quality of life and decrease pain and other symptoms of fibromyalgia [9,10,11]. Recent preclinical studies are currently investigating a beneficial impact on the resolution of this disease through different approaches. Among the most current is the use of a new compound called Hydrox® (HD), which contains 40–50% hydroxytyrosol. The results show that this supplement could activate the WNT/catenin signaling route after reserpine-induced FM [12].

On the other hand, an important factor related to pain and FM is oxidative stress through the production of reactive oxygen species (ROS). These are formed by oxidation at low levels in the body′s cells and tissues and their concentration is controlled by the action of a defense system made up of enzymes and non-enzymatic species [8]. However, if oxidation levels increase due to disposal complications or other circumstances, it leads to significant oxidative stress that can cause metabolic and biological macromolecule alterations [13]. Such stress can lead to peripheral and central sensitization and affect nociception. This interferes with the musculature through a decrease in nociceptors locally, resulting in a decreased pain threshold with pain as a characteristic symptom [14]. In addition, the increase in oxidative stress in these patients is increasing and is related to the severity of symptoms. Therefore, the relationship between these symptoms and an imbalance between oxidation products and antioxidant defenses has been established [15].

Although oxidative stress is thought to play an important role in the pathogenesis of FM, further studies are needed [16]. One of the ways to counteract the excess of free radicals is to resort to certain nutrients, such as antioxidants. The higher the level of antioxidants in our body, the more protected we are against oxidative damage. In people with FM, a decrease in antioxidant levels can increase pain [17].

Therefore, antioxidants such as vitamins, coenzyme Q10, virgin olive oil, and alpha-lipoic acid (ALA) are of interest due to their association with the characteristic symptoms of FM, one of the main symptoms being pain [18,19,20,21,22]. Other mineral supplements, such as magnesium or iron, could be used as a co-treatment in this disease, helping to counteract the level of pain and improve quality of life [23]. This is because FM patients with reduced magnesium levels are related to low-grade swelling, muscle weakness, and paresthesia, all of which are common symptoms of FM [24]. In the case of iron, depletion leads to reduced production of biogenic amines [25]. The beneficial effects of vitamin D, as an antioxidant, on pain and its possible association with FM have already been highlighted in a previous review [26], although we note that there is no consensus on the association between vitamin D and FM specifically. However, a correlation between low vitamin D status and non-specific musculoskeletal pain has been demonstrated [27].

Given the wide range of antioxidant supplements used to treat pain caused by FM, as well as the great heterogeneity in the duration of these treatments, the following systematic review is proposed to determine the possible beneficial effects of antioxidant supplementation on pain levels perceived by patients diagnosed with FM. We also aim to determine the best duration of treatment to reduce pain in FM patients.

## 2. Materials and Methods

### 2.1. Review Procedure

For this review, we used the guidelines set out in the protocol of preferred reporting items for systematic reviews and meta-analyses (PRISMA) [28]. To select studies, the PICOS question model [29] was used as follows: P (population), “people with FMS”; I (intervention), “administration or supplementation of antioxidant substances”; C (comparison), “some conditions without supplementation”; O (outcomes), “changes in perceived pain level across different pain questionnaires in fibromyalgia”. The types of pain outcomes were to be assessed by validated pain scales in FMS patients. The most frequent being Pain Catastrophizing Scale (PCS), Visual Analogue Scale (VAS), Chronic Pain Grading Scale (CPGS), Short Form McGill Pain Questionnaire (SF-MPQ), Fibromyalgia Impact Questionnaire (FIQ), Brief Pain Inventory (BPI), Present Pain Intensity (PPI), and Pain Pressure Threshold (PPT) [30,31,32,33,34,35].

A structured search of the following databases was performed: Medline (PubMed), Scopus, Web of Science (WOS), Cochrane Database of Systematic Reviews, and Cochrane Central Register of Controlled Trials. The controlled medical vocabulary used was that set out in MeSH and keywords related to FMS, antioxidants, and pain. They were as follows: (“fibromyalgia” OR “fibromyalgia syndrome”) AND (“antioxidants” OR “supplements”) AND (“pain”). A snowball strategy was used to find articles related to the topic under review. Titles and abstracts of identified studies were cross-checked to discard duplicates and unrelated studies. After selection of those related to the review we proceeded to read them in full. The search for published studies was conducted independently by two authors and disagreements were resolved by assessment by a third author.

### 2.2. Process Followed for the Selection of Articles

Aspects related to age and race/ethnicity were not filtered for selection. Inclusion criteria focused on: (i) showing an adequate design in the use of antioxidant supplements in humans; (ii) the use of antioxidants even if they are of different types due to the difficulty of finding different studies with the same type of antioxidant; (iii) randomized controlled study with or without placebo; and (iv) specific information related to the pain variable and the validated scales for fibromyalgia mentioned above. Exclusion criteria included (i) studies with no human outcomes and not related to pain; (ii) animal studies; (iii) studies with other types of interventions or diets mixed with nutritional supplements other than antioxidants (if the study contains many types of antioxidants as a mixture, it is uncertain which one acts as a beneficial effect or not); (iv) studies published in languages other than English or Spanish; (v) studies using vitamin D for its previous scientific evidence of benefit for pain; and (vi) editorials, review studies, and letters to the editor.

### 2.3. Data Extraction

Once the criteria established for the selection of each study had been applied, we proceeded to extract the data corresponding to the relevant aspects of our study, which are shown under the heading of the tables in this review. For this purpose, a spreadsheet was used by two researchers and then jointly agreed with a third researcher when there were discrepancies.

### 2.4. Methodological Quality

The review of the methodology of the selected studies was carried out using the McMaster critical review form for quantitative studies [36]. Studies that did not meet the methodological quality requirements were excluded. This review was carried out by the same researchers as above and discrepancies were also resolved in the same way with the participation of the third researcher.

## 3. Results

### 3.1. Selection of Studies

The review initially extracted 277 records, of which 150 were eliminated due to duplication, leaving a total of 127 articles. Likewise, after eliminating studies which did not meet the inclusion criteria (*n* = 49), 78 records were selected. Subsequently, 38 studies were eliminated because 7 were conducted in animals, 9 were non-Spanish or non-English, 6 were published as letters to the editor or literature reviews, and 16 were not related to food or supplements with antioxidant characteristics. Of the 40 full-text articles that remained for eligibility, 23 studies were excluded due to an inadequate overall study protocol (*n* = 6) and lack of an appropriate control design for supplementation (*n* = 17). The remaining 17 studies were included in this review (Figure 1).

### 3.2. Results of the Quality Assessment

Then we made the quality assessment of each selected article. The score of the evaluated articles ranged from 12 to 16 points. Two studies were assessed as “excellent,” twelve as “very good,” and three as “good.” The results with the scores for each study are presented below (Table 1).

### 3.3. Descriptive Information of the Selected Articles Included in the Systematic Review

This systematic review included 17 experimental studies. The characteristics of each study necessary to obtain relevant data in relation to our objective are displayed in Table 2. Table 3 shows the synthesis of the studies included in the systematic review with the intervention and pain measurement instruments.

Nine studies reported benefits in reducing pain scores [37,39,40,43,44,45,47,50,51]. In the remaining eight studies there was no evidence of benefit after the intervention [22,38,41,42,46,48,49,52].

Table 3 shows that all studies that have used coenzyme Q10 as an antioxidant at a dose of at least 300 mg/day and for between 10 [39] and 24 weeks [43] have shown beneficial effects on pain reduction. We can also note that these studies also agree that they all have in common the use of the VAS scale in the measurement of pain [39,43,47]. Two studies show benefits [37,44] using vitamins with a minimum daily dose of 200 mg, with a duration of at least 6 weeks of supplementation and agreeing on the use of the VAS scale to measure pain perception after supplementation. The rest of the studies with beneficial effects have used minerals and algae with high antioxidant power [40,45,51], with all but one [40] coinciding in the use of the VAS scale as a measure of pain perception after supplementation and in having supplemented for more than 6 weeks.

## 4. Discussion

This review summarizes the possible relationship between FM and antioxidant supplements and their relationship to possible effects on pain reduction. Pain is one of the conditions that can influence and alter oxidative stress. For example, a disequilibrium between pro-oxidants and antioxidants in people with fibromyalgia and persistent pain suggests that there may be an influence on nociceptive processing. [52,53].

A total of 17 studies analyzed specifically assessed the pain symptom using one of the eight scales used for pain in the different trials. Perception of pain as scored on the scales improved significantly after consumption of some antioxidants, as discussed in the previous section. The most commonly used scales for measuring pain perception were VAS (76.47%) and FIQ (47.05%). The study of bibliography has shown that the implication of antioxidants supplements is not without controversy, although clinical trials with CoQ10 and alpha-lipoic acid (ALA) show promising results [23,54], according to this review. In the present discussion, we present each of the antioxidants reviewed in the trials under study and discuss their benefits and possible controversies and evidence.


*Alpha-Lipoic Acid (ALA) and Acetyl-L-Carnitine (LAC)*


There is strong evidence that ALA and LAC are effective for peripheral neuropathy, especially in diabetics [22,55]. They not only cut down pain but also improve numbness and tingling. There is evidence to suggest possible benefits in reducing the frequency and severity of migraines and pain associated with fibromyalgia. However, this evidence needs further support [56]. In the study by Gilron, et al. (2021) [22] there is no evidence of a significant effect to demonstrate this, so we will have to wait for further studies to definitively prove this or not. One aspect observed in this review related to this type of antioxidant is the need for more RCTs that can confirm sufficient benefits in practical application and be extrapolated to a larger number of patients worldwide.

However, in this review we have shown that ALA is beneficial in reducing pain perception in FM. Compared with placebo, VAS scores improved significantly after ALA supplementation (*p* < 0.05) [49], although we could not find specific scores.


*Coenzyme Q10*


Ubiquinone plays a key role in oxidative phosphorylation. Coenzyme Q10 has been shown to beneficially stimulate the AMPK gene, which may be responsible for the inflammation, low antioxidant levels, and low mitochondrial production that characterize the pathophysiology of fibromyalgia [57]. This review shows that there are studies demonstrating its benefits for pain [38,42,46]. The pathophysiology of FM may be influenced by oxidative stress by detecting reduced levels of coenzyme Q10 in blood mononuclear cells derived from FM patients [58]. The study by Miyamae, et al. (2013) [59] showed that CoQ10 levels are reduced in these patients, but that supplementation can restore levels and reduce fibromyalgia symptoms, including pain. However, this study has not been included in our review because it did not meet the inclusion criteria, as it was conducted in children. This research supports the results discussed in this review.

Importantly, supplementation with CoQ10 and pregabalin provides additional benefit in relieving pain sensation in patients with FM [38]. In addition, the study by Pierro, et al. (2016) [42] also confirms the beneficial effects of CoQ10 in counteracting pain in women affected by fibromyalgia. It shows that, compared with a control group, CoQ10 administration significantly improved most pain-related outcomes by 24–37%. However, it does not have sufficient statistical evidence due to the limitation pointed out by the authors themselves on the limited number of participants. Overall, Q10 supplementation in the three studies [38,42,46] included in this review did not differ greatly in terms of the amount of CoQ10 300–400 mg/day administered. However, in terms of administration time, one of them [42] doubled the administration time compared with the other two. In addition to the need for a larger number of participants, we also highlight the need for a longer CoQ10 administration time for comparison.


*Vitamins*


Randomized placebo-controlled trials can show the therapeutic role of vitamin C, acerola root, and freeze-dried royal jelly [36], and vitamin E, vitamin C and Nigella Sativa [43]. Both studies showed benefits in decreasing pain perception after supplementation; however, on different measurement scales. Pain perception showed a decrease in pain perception with the VAS-Pain scale in the study by Iqbal, et al. (*p* < 0.05) [44] and with the FIQ scale in the study by Bermarki, et al. [36].

This study compared both the efficacy and safety of a supplement called FibromyalgineR (Fib) (vitamin C, acerola root, and freeze-dried royal jelly) with that of another food supplement (FS) (acting as a placebo) and with a control arm that received no supplement. The Fib vitamin supplement resulted in a significant improvement (*p* < 0.001) relative to the other two study groups on the FIQ scale only. It is important to note that the FIQ scale only measures pain intensity on item five, and the VAS scale is specific to pain only; we cannot claim a significant reduction by this supplement in overall pain rating, although it does improve pain intensity. However, another study has also been reported that showed a decrease in VAS pain in FM patients treated with an improved diet and vitamin supplementation in an open-label, non-randomized controlled study [60]. These results confirm existing evidence that supplementation with antioxidants such as vitamins C and E to therapy may be helpful in treating FM symptoms. Ginger, which is a potent antioxidant, may also act on fatigue and pain by decreasing oxidative stress [61].

In addition, vitamins such as vitamin C have analgesic effects on pain as demonstrated by clinical trials. [62].


*Other Types of Antioxidants*


Other types of antioxidants used in different trials analyzed showed neutral effects on the reduction of pain perception in FM patients. These antioxidants were turmeric supplementation [37], EVOO [40], caffeine [41], creatine monohydrate [45], soy protein with soy isoflavone [47], cherry juice [48], and malic acid with magnesium hydroxide [51]. However, we note that in the study by Bagis, et al. [44], combining magnesium citrate with amitriptyline did show beneficial effects in reducing pain perception using the same VAS pain scale. Magnesium citrate supplementation in combination with amitriptyline was effective in reducing pain, intensity, and other fibromyalgia-related parameters. However, uncombined magnesium citrate only had an effect on pain points. [44]. These are important data for future studies.

Among these antioxidants, we focus specifically on two due to their extensive use in food and in different studies. These are extra virgin olive oil (EVOO) and turmeric. The effects of olive oil intake on cardiovascular disease [63] and rheumatoid arthritis [64] have been beneficial. Both coincide in being diseases with an association with oxidative stress. However, few studies are available that measure the effect of this type of antioxidant in women with FM in relation to pain [40].

The antioxidant activity of EVOO is responsible for the protection of DNA, proteins, and lipids against ROS, and it is in FM that several studies found elevated levels of ROS [23,40]. The clinical trial reviewed in this study investigated the effect of 50 mL/day EVOO compared to refined olive oil in 23 female subjects with FM. Comparing extra virgin olive oil with refined olive oil after 21 days of intervention, one study showed that protein charring and lipid peroxidation were significantly improved. However, there was no improvement in the pain variable [40]. Therefore, although its efficacy on pain is promising, more studies with this type of antioxidant are needed.

The other antioxidant widely used in everyday life as a spice is turmeric. Turmeric is a spice that has antioxidant, anti-inflammatory, antiviral, and antifungal properties [65]. The effects of a turmeric-based supplement in women with fibromyalgia have not shown a beneficial effect on scales of perceived degree of chronic pain in FM patients [37].

Finally, we conclude that more scientific evidence is needed to show whether turmeric could actually improve chronic pain in FM. Several human studies have found some evidence for the anti-inflammatory activity of curcumin [65]. However, no statistically significant benefit in reducing perceived pain in FM patients has been demonstrated, although it may be recommended for its general anti-inflammatory benefits [66,67].

### 4.1. Study Limitation

There are several limitations to note about this systematic review. The data from the studies could not be pooled due to methodological diversity, and there is heterogeneity among the studies, mainly with the time of intervention and the type of antioxidants. We suggest increasing the time and sample with antioxidants that require more scientific evidence to affirm their beneficial effect on pain perception.

The scales used to measure pain perception levels were different in four studies, the rest all used at least the VAS scale, which is the most commonly used.

In terms of the limitations of the studies analyzed, we highlight the following. Firstly, the sample size is rather small in the published studies. Secondly, the methodology used to measure the results is very heterogeneous and does not take into account the confounding phenomenon.

Nevertheless, a decrease in fibromyalgia pain was observed in studies using coenzyme Q10, acetyl-l-carnitine, combination of vitamin C, E, and Nigella sativa seeds, vitamin C, acerola ginger root, freeze-dried royal jelly, ferric carboxymaltose, and a combination of amitriptyline + magnesium citrate and Chlorella green algae.

### 4.2. Recommendations for Further Research 

Further research is needed with larger sample sizes and using homogeneous measurement scales that can have more scientific rigor. In addition, oxidative stress levels should be measured and inflammatory biomarkers should be analyzed with antioxidants to understand their influence on FM pathophysiology.

Research combining substances based on antioxidant and anti-inflammatory mechanisms is necessary; however, we believe that we first need more separate research to understand further combined modality research to understand how it may or may not influence FM. In the future these studies could establish an active and targeted treatment for pain in these patients.

## 5. Conclusions

This review encompassed the literature and showed that the role of antioxidant supplements in FM could improve pain perception in patients, as measured by scales for this purpose.

Nine studies showed a significant improvement in pain perceived by FM patients that used the VAS scale and/or the FIQ scale. Specifically, supplementation with Fibromyalgine® (Fib) (vitamin C, acerola ginger root, and freeze-dried royal jelly), coenzyme Q10 alone in combination with Pregabalin, ferric carboxymaltose, vitamin C, E, and Nigella sativa, magnesium + amitriptyline, LAC, and Sun Chlorella™ green algae appear to be effective in reducing FM pain perception.

In addition, supplementation time could be associated with improved quality of life in these patients. Suggesting that supplementation time could have a functional impact on the efficacy of antioxidants on pain.

These results should be interpreted with caution due to the limitations mentioned above. These limitations have not allowed us to make solid comparisons due to the heterogeneity of the antioxidants used and the short supplementation times carried out.

Finally, we believe that there is a lack of studies that analyze the duration of the beneficial effect of antioxidants on pain.

## Figures and Tables

**Figure 1 jcm-11-02462-f001:**
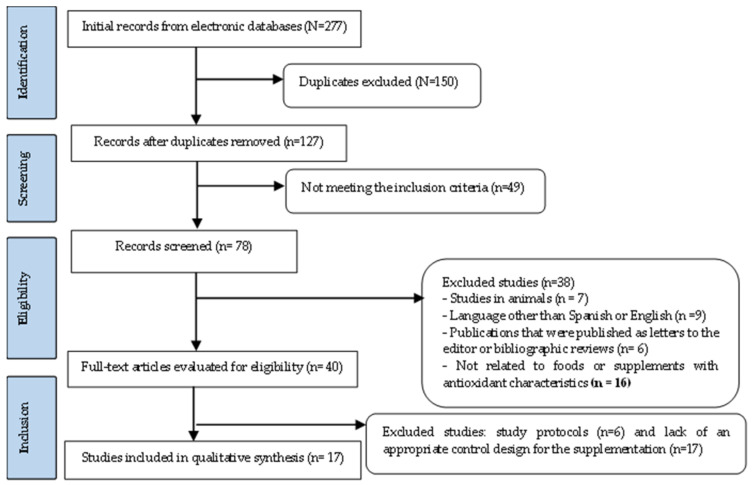
Flowchart for item selection.

**Table 1 jcm-11-02462-t001:** Quality assessment of the studies included in the systematic review by McMaster critical review form for quantitative studies [36].

Author/s	Items according to Critical Review By McMaster																T1	%	MQ
	1	2	3	4	5	6	7	8	9	10	11	12	13	14	15	16			
Gilron I, et al., 2021	1	1	1	1	1	1	1	0	1	1	1	1	1	1	1	0	14	87.5	VG
Barmaki M, et al., 2019	1	1	1	1	0	1	0	1	1	1	1	0	1	1	1	0	12	75	G
San Mauro I, et al., 2019	1	1	1	1	1	0	0	0	1	1	1	1	1	1	1	1	13	81.25	VG
Sawaddiruk P, et al., 2019	1	1	1	1	1	1	1	1	1	1	1	0	1	1	1	0	14	87.5	VG
Boomershine, et al., 2018	1	1	1	1	1	1	1	1	1	1	1	1	1	0	1	1	15	93.75	E
Rus A, et al., 2017	1	1	1	1	0	1	0	0	1	1	1	1	1	1	1	1	13	81.25	VG
Umeda M, et al., 2016	1	1	1	1	1	1	1	0	1	1	1	1	1	1	1	0	14	87.5	VG
Di Pierro F, et al., 2016	1	1	1	1	1	1	0	1	1	1	1	0	1	1	1	0	13	81.25	VG
Iqbaq R, et al., 2015	1	1	1	1	0	1	0	1	1	1	1	1	1	1	1	1	14	87.5	VG
Bagis, et al., 2013	1	1	1	1	1	0	1	1	1	1	1	1	1	0	1	1	14	87.5	VG
Alves, et al., 2013	1	1	1	1	0	1	0	0	1	1	1	1	1	1	1	1	13	81.25	VG
Cordero, et al., 2012	1	1	1	1	0	1	1	0	1	1	1	0	1	1	1	1	13	81.25	VG
Wahner-Roedler, et al., 2011	1	1	1	1	1	1	1	1	1	1	1	0	1	1	1	0	14	87.5	VG
Elliot D, et al., 2010	1	1	1	1	1	1	1	1	1	1	1	1	1	1	1	0	15	93.75	E
Rossini, et al., 2007	1	1	1	1	0	1	0	0	1	1	1	1	1	1	1	1	13	81.25	VG
Merchant, et al., 2001	1	1	1	1	0	1	0	1	1	1	1	0	1	1	1	0	12	75	G
Russel, et al., 1995	1	1	1	1	0	0	0	1	1	1	1	0	1	1	1	1	12	75	G
T2 + A1:S14	19	19	19	19	11	15	9	11	19	19	19	11	19	17	19	10			

Abbreviations: (1) Criterion was met; (0) Criterion was not met; (T1) Total items fulfilled by study; (T2) Number of studies that fulfilled the item; (%) Percentage of methodological quality assessment; (MQ) Methodological quality; (A) acceptable 9–10 points; (G) good 11–12 points; (VG) very good 13–14 points; (E) excellent ≥ 15 points.

**Table 2 jcm-11-02462-t002:** Synthesis of the studies included in the systematic review with type of study, participants, and groups.

Author/s, Year of Study	Country	Participants, Sex	Age (±)	Intervention	Placebo	Study Duration
Gilron I, et al., 2021 [22]	Canada	*n* = 22 women, 5 men	47 ± 6.72	27	-	10 weeks *^G^
Barmaki M, et al., 2019 [37]	France	*n* = 100 all women	49 ± 7.12	I1 = 36; I2 =33	31	24 weeks
San Mauro I, et al., 2019 [38]	Spain	*n* = 13 all women	51.46 ± 8.04	6	7	4 weeks
Sawaddiruk P, et al., 2019 [39]	Thailand	*n* = 2 men, 9 women	46 ± 11	5	6	10 weeks
Boomershine, et al., 2018 [40]	US	*n* = 80 women, 1 man	42,5 ± 10.9	41	40	6 weeks
Rus A, et al., 2017 [41]	Spain	*n* = 23 all women	50.88 ± 6.5	11	12	3 weeks
Umeda M, et al., 2016 [42]	US	*n* = 23 all women	43.57 ± 18.49	12	11	3 sessions
Di Pierro F, et al., 2016 [43]	Italy	*n* = 22 all women	53 ± 9.1	12	10	24 weeks
Iqbaq R, et al., 2015 [44]	Pakistan	*n* = 50 all woman	37.87 ± 1.68	50	16	8 weeks
Bagis, et al., 2013 [45]	Turkey	*n* = 80 all women	41.4 ± 10.5	60	20 **	8 weeks
Alves, et al., 2013 [46]	Brazil	*n* = 28 all women	48.85 ± 9.25	15	13	16 weeks
Cordero, et al., 2012 [47]	Spain	*n* = 35 all women	45.75 ± 4.5	20	15 **	12 weeks
Wahner-Roedler, et al., 2011 [48]	US	*n* = 50 all woman	47.7 ± 4.25	25	25	6 weeks
Elliot D, et al., 2010 [49]	US	*n* = 14 all women	51 ± 2.0	14	-	4 weeks *^E^
Rossini, et al., 2007 [50]	Italy	*n* = 89 all women	46.8 ± 5.05	47	42	10 weeks
Merchant, et al., 2001 [51]	US	*n* = 43 all women	47.1 ± 9	22	21	12 weeks
Russel, et al., 1995 [52]	US	*n* = 21 women, 3 men	49	12	12	4 weeks

Notes: * Crossover study (*^G^ Each period lasted 5 weeks, with a 4-week treatment period and a 1-week washout period; *^E^ Each period lasted 2 weeks, with a 10-day treatment a 4-day washout period) ** Healthy controls had no signs or symptoms of FM.

**Table 3 jcm-11-02462-t003:** Synthesis of the studies included in the systematic review with types of supplementation and doses and instruments for measuring pain.

Study	Intervention Group (IG)	Control	Pain Scale	Results	Effect for Pain
Gilron I, et al., 2021 [22]	IG = ALA * 1663 mg/day for 5 weeks and placebo during the second 5 weeks. IGP = the first 5 weeks were treated with placebo and ALA for the next 5 weeks.	Placebo	FIQ, BPI, VAS	For women, the perception of pain for all scales with respect to the placebo group was for ALA of (*p* = 0.13) and for men (*p* = 0.01).	Neutral effect for women and beneficial for men.
Barmaki M, et al., 2019 [37]	G1 = Fibromyalgine® (Fib) (vitamin C, acerola ginger root, freeze-dried royal jelly), 2 capsules/day; G2 = food supplement (FS), 2 capsules/day; G3 = control arm not receiving any supplementation.	NoST	FIQ, VAS	The supplementation with Fibromyalgine® showed an improvement in pain intensity on the FIQ scale (*p* < 0.001).	Positive benefit.
San Mauro I, et al., 2019 [38]	Turmeric supplement 500 mg/day, gluten-free diet and low in histamine.	NoST	CPGS, PCS	PCS (*p* = 0.190), GPGS (*p* = 0.671).	Neutral benefit.
Sawaddiruk P, et al., 2019 [39]	G1 = CoQ10 supplementation 300 mg/day+ pregabalin (150 mg/day); G2 = placebo + pregabalin (150 mg/day) for 40 days. At day 40, patients who received CoQ10 therapy were switched to placebo, and vice versa.	Placebo	VAS, PPT	Decrease in VAS and increase in PPT significantly increased in pregabalin-treated FM patients with CoQ10, compared to those treated with pregabalin and placebo alone.	Positive benefit.
Boomershine, et al., 2018 [40]	Ferric carboxymaltose 15 mg/kg (up to 750 mg).	Placebo	FIQ, BPI	Greater improvements from baseline to day 42 were observed for ferric carboxymaltose vs. placebo in FIQ total score and BPI total score.	Positive benefit.
Rus A, et al., 2017 [41]	Extra virgin olive oil (EVOO) 50 mL/day. Control group = Refined olive oil (ROO) 50 mL/day.	Control	FIQ, VAS	In the EVOO group, a decrease in FIQ (*p* < 0.011) was observed, but not in pain (*p* < 0.279) compared to the ROO consumption group.	Neutral benefit.
Umeda M, et al., 2016 [42]	Gum with 100 mg of caffeine.	Placebo	SF-MPQ, PPI, VAS	Pain results improved in the experimental group measured with SF-MPQ (*p* = 0.006). Pain measured with VAS (*p* = 0.396)/PPI (*p* = 0. 87).	Neutral benefit.
Di Pierro F, et al., 2016 [43]	200 mg × 2/day CoQ10 formula.	Control	VAS, FIQ	Statistical significance is only evidenced *p* <0.005, for the pain scale (VAS), in the rest the results were not significant.	Positive benefit.
Iqbaq, et al., 2015 [44]	Vitamin C (200 mg daily), E (200 mg daily) and Nigella sativa seeds (13 mg 4–5 times daily).	Control	VAS	VAS (*p* < 0.05).	Positive benefit.
Bagis, et al., 2013 [45]	IG1 (n = 20) Mg citrate 300 mg/day; IG2 (n = 20) amitriptyline 10 mg/day; GI3 (n = 20) Mg citrate 300 mg/day + amitriptyline 10 mg/day.	NA	VAS, FIQ	Positive effects on all pain parameters with the combination of amitriptyline + magnesium citrate proved to be effective on all pain parameters (*p* < 0.001).	Positive effects combination of amitriptyline + magnesium citrate.
Alves, et al., 2013 [46]	20 gm of creatine monohydrate for 5 days divided into 4 equal doses, followed by 5 gm/day as a single dosage throughout the trial.	Placebo	MPQ, FIQ	FIQ and MPQ (*p* > 0.005).	Neutral benefit.
Cordero, et al., 2012 [47]	300 mg/day of CoQ10.	Control	FIQ, VAS	There was a decrease in the FIQ score (*p* < 0.001) and VAS (*p* < 0.001) in the experimental group compared with the control group.	Positive benefit.
Wahner-Roedler, et al., 2011 [48]	Soy protein (20 g), soy isoflavone (160 mg) (1 serving daily).	Placebo	FIQ	No significant differences between groups.	Neutral benefit.
Elliot D, et al., 2010 [49]	GE = received tart cherry juice, 2 bottles/day, morning and evening.	Placebo	VAS	There were no significant differences for either group in terms of pain (*p* > 0.005).	Neutral benefit.
Rossini, et al., 2007 [50]	1000 mg acetyl L-carnitine (LAC) or placebo.	Placebo	VAS	VAS (*p* < 0.02).	Positive benefit.
Merchant, et al., 2001 [51]	Sun Chlorella™ green algae and Wakasa Gold Chlorella™ (500 g and 100 mL/day, respectively)	Placebo	VAS	VAS (*p* < 0.05).	Positive benefit.
Russel, et al., 1995 [52]	200 mg malic acid + 50 mg magnesium, from 3 capsules up to 6 per day.	Placebo	VAS	Pain with this supplement was not significantly different from placebo (P5 0.7).	Neutral benefit.

Notes: * Alpha-Lipoic Acid Capsules (ALA); Fibromyalgia Impact Questionnaire (FIQ), Brief Pain Inventory (BPI), Visual Analogue Scale (VAS), Chronic Pain Grade Scale (CPGS), Pain Catastrophizing Scale (PCS), Pain Pressure Threshold (PPT), Short-form McGill Pain Questionnaire (SF-MPQ), Present Pain Intensity (PPI). Supplementary treatment (NoST); NA (not applicable); Positive = significant improvement in pain perception; Neutral = no significant improvement in pain perception.

## Data Availability

Not applicable.

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
