# Peer review of "Effects of Antioxidants on Pain Perception in Patients with Fibromyalgia—A Systematic Review"

_jcm, 2022, doi:10.3390/jcm11092462_

Round 1

Reviewer 1 Report

The manuscript of ... is interesting and well structured. However, minor changes are required:

  • authors should update references;
  • it would be interesting to insert a paragraph on recent preclinical studies that could have an important impact on the resolution of the disease;
  • Please refer to doi: doi.org/10.3390/ijms21217877; 

    doi.org/10.3390/biomedicines9111683; doi: 10.5772/intechopen.70016; doi: 10.3390/ijms22083891

  • The authors should better check the manuscript for any typographical errors

Author Response

We would like to sincerely thank for your helpful recommendations. We have seriously considered all the comments and carefully revised the manuscript accordingly. Revisions are highlighted in yellow through the manuscript to indicate where changes have taken place. We feel that the quality of the manuscript has been significantly improved with these modifications and improvements based on the reviewers’ suggestions and comments. We hope our revision will lead to an acceptance of our manuscript for publication in Journal of Clinical Medicine.

Response to Reviewer 1 Comments

  1. Indeed, these articles support current research and are necessary to better support our article.

1: Unfortunately, conventional medical therapies targeting this pathology produce limited benefits. Review studies suggest that the combination of pharmacological and alternative therapies (including heat and light treatments, the use of bioactive plant molecules, electro stimulators and body exercises) could improve quality of life and reduce pain and other FM-related symptoms [9,10,11].

Recent preclinical studies are currently investigating a beneficial impact on the resolution of this disease through different approaches. Among the most current is the use of a new compound called Hydrox® (HD), which contains 40-50% hydroxytyrosol. Results showed that HD was able to modulate the activation of the WNT/catenin signalling pathway following reserpine-induced FM [12].

Please refer to doi:

9     Maffei, M.E. Fibromyalgia: Recent Advances in Diagnosis, Classification, Pharmacotherapy and Alternative Remedies. Int J Mol Sci. 2020, 23,21(21):7877. doi: 10.3390/ijms21217877.

10   Siracusa, R.; Paola, R.D.; Cuzzocrea, S.; Impellizzeri, D. Fibromyalgia: Pathogenesis, Mechanisms, Diagnosis and Treatment Options Update. Int J Mol Sci. 2021 Apr 9;22(8):3891. doi: 10.3390/ijms22083891.

11   Brito, R.G.; Santos, P.L.; Almeida, M.; Pina, L.T.S.; Antoniolli, A.R.; Almeida, J.R.G.d.S.; Picot, L.; Gokhan, G.; Quintans, J.S.S.; Júnior, L.J. Natural Products as Promising Pharmacological Tools for the Management of Fibromyalgia Symptoms – A Review. In (Ed.), Discussions of Unusual Topics in Fibromyalgia. 2017. IntechOpen. https://doi.org/10.5772/intechopen.70016

12   D'Amico, R.; Cordaro, M.; Siracusa, R.; Impellizzeri, D.; Trovato, A.; Scuto, M.; Ontario, M.L.; Crea, R.; Cuzzocrea, S.; Di Paola, R.; Fusco, R.; Calabrese, V. Wnt/β-Catenin Pathway in Experimental Model of Fibromyalgia: Role of Hidrox®. Biomedicines. 2021,13;9(11):1683. doi: 10.3390/biomedicines9111683.

  1. The authors should better check the manuscript for any typographical errors

2: We have reviewed and corrected spelling errors, thank you for your appreciation. And we have also gone on to review with a native English speaker.

Reviewer 2 Report

This review considers studies of the outcomes on pain in patients with fibromyalgia with antioxidant supplements. This is a very topical subject however several aspects of the manuscript need attention.

The abstract is misleading of the effectiveness of antioxidant supplements. The statement ‘Our review found the effectiveness of antioxidant supplementation to reduce pain…..’, needs to be modified to emphasize ‘…..to reduce pain in only 53% of the studies…’.

In the introduction, penultimate paragraph’ the statement ‘….and other such magnesium or iron supplements…’ is lacking clarity and needs revising to enable the reader to understand the importance.

How does ‘…40% of FM subjects … reported with vitamin D deficiency..’ compare with the proportion of the general population that do not develop pain with vitamin D deficiency? If the proportion is similar or greater in the general population and they have not developed pain, what is the relevance of vitamin D deficiency to the pathophysiology of FM?

The final paragraph of the introduction is not grammatically correct and needs revising.

In the methods section 2.3 the sentence ‘Therefore, inclusion criteria included articles (i) depicting a well-designed 104 study that included the use of that included supplementation with antioxidants’ is clumsy and confused and needs revision.

In the results section 3.1 ‘We also excluded 127 articles after full-text review.’ Is not consistent with the content of Figure 1. Please revise.

Table 1 needs an explanation of the identity of Items 1-16, especially for those readers not familiar with this methodology.

Section 3.3 ‘This systematic review included ten experimental studies’ needs to be corrected to 17. The statement ‘All eight studies remained neutral benefits with a non-significant marginal benefit in a subset of patients after the intervention’ is a meaningless and misleading statement and needs to be removed. In the remaining 8 studies there was no evidence of benefit. The structure of text in the Final paragraph is very confused which limits understanding of the meaning, this requires revision.

Table 2. The total number of participants and number of participants per arm have a number of inconsistencies within the table and with respect to some of the publications. All of the data requires confirmation and revision where appropriate. In addition, ‘Duration’ needs to be changed to ‘Duration of Study’ and where appropriate (eg crossover studies) is different from Duration of Treatment.

Since many aspects of lifestyle, eg diet, exercise, can influence antioxidant status of a persons biology, it would be beneficial to the reader to be informed of the inclusion and exclusion criteria applied to recruitment of participants in all the studies.

In table 3 change increasement and decreasement with increase and decrease.

In the discussion of Coenzyme Q10, the last paragraph should provide some consideration of the observation suggesting in the studies of short duration pain and FIQ scores were significantly improved but the study of longer duration only pain was reduced. This could be consistent with beneficial effects waning rather than being maintained with continued use of Coenzyme Q10.

In the conclusion, it needs to be emphasized that significant improvement was only observed in a small majority, ie 53%, of studies so that the reader is not mislead with respect of the level of effectiveness demonstrated of antioxidant supplementation. The suggested ‘minimum time of a month and a half’, how was this period derived or calculated?

Author Response

We would like to sincerely thank for your helpful recommendations. We have seriously considered all the comments and carefully revised the manuscript accordingly. Revisions are highlighted in yellow through the manuscript to indicate where changes have taken place. We feel that the quality of the manuscript has been significantly improved with these modifications and improvements based on the reviewers’ suggestions and comments. We hope our revision will lead to an acceptance of our manuscript for publication in Journal of Clinical Medicine.

Response to Reviewer 2 Comments

  1. The abstract is misleading of the effectiveness of antioxidant supplements. The statement ‘Our review found the effectiveness of antioxidant supplementation to reduce pain…..’, needs to be modified to emphasize ‘…..to reduce pain in only 53% of the studies…’.

1: This review found that antioxidant supplementation is efficient in reducing pain in nine of the studies reviewed. Studies with a duration of supplementation of at least six weeks showed a benefit on pain perception in 80% of the patients included in these studies. The benefits shown by vitamins and coenzyme Q10 are remarkable.

  1. In the introduction, penultimate paragraph’ the statement ‘….and other such magnesium or iron supplements…’ is lacking clarity and needs revising to enable the reader to understand the importance.

2: Other mineral supplements, such as magnesium or iron, could be used as a co-treatment in this disease, helping to counteract the level of pain and improve quality of life [23]. This is because FM patients with reduced magnesium levels are associated with low-grade swelling, muscle weakness and paraesthesia, which are typical symptoms of FM [24]. And in the case of iron, iron depletion leads to reduce production of biogenic amines [25]. Given the wide range of antioxidant supplements used to treat pain caused by FM, as well as the great heterogeneity in the duration of these treatments, the following systematic review is proposed to determine the possible beneficial effects of antioxidant supplementation on pain levels perceived by patients diagnosed with FM. We also aim to determine the best duration of treatment to reduce pain in FM patients.

  1. How does ‘…40% of FM subjects … reported with vitamin D deficiency..’ compare with the proportion of the general population that do not develop pain with vitamin D deficiency? If the proportion is similar or greater in the general population and they have not developed pain, what is the relevance of vitamin D deficiency to the pathophysiology of FM?

(In relation to this contribution, it is true that we cannot say that vitamin D acts solely and exclusively on the pathophysiology of FM, but it does act on pain in general. We have changed the wording and set out the current reality, which is better explained)

3: The beneficial effects of vitamin D, as an antioxidant, on pain and its possible association with FM have already been highlighted in a previous review [26]. Although we note that there is no consensus on the association between vitamin D and FM specifically. However, a correlation between low vitamin D status and non-specific musculoskeletal pain has been demonstrated [27].

  1. The final paragraph of the introduction is not grammatically correct and needs revising.

4: Given the wide range of antioxidant supplements used to treat pain caused by FM, as well as the great heterogeneity in the duration of these treatments, the following systematic review is proposed to determine the possible beneficial effects of antioxidant supplementation on pain levels perceived by patients diagnosed with FM. We also aim to determine the best duration of treatment to reduce pain in FM patients.

  1. In the methods section 2.3 the sentence ‘Therefore, inclusion criteria included articles (i) depicting a well-designed 104 study that included the use of that included supplementation with antioxidants’ is clumsy and confused and needs review.

5: (i) showing a well-designed study involving the use of antioxidant supplements in humans;

  1. In the results section 3.1 ‘We also excluded 127 articles after full-text review.’ Is not consistent with the content of Figure 1. Please revise.

6: … leaving a total of 127 articles. Likewise, after eliminating articles that did not meet the inclusion criteria (n=49), 78 records were selected. Subsequently, 38 studies were eliminated because 7 articles were conducted in animals, 9 additional non-Spanish or non-English studies, 6 publications that were published as letters to the editor or literature reviews, and 16 not related to food or supplements with antioxidant characteristics. Of the 40 full-text articles that remained for eligibility, 23 studies were excluded due to an inadequate overall study protocol (n=6) and lack of an appropriate control design for supplementation (n=17). The remaining 17 studies met the inclusion criteria and were included in the present systematic review (Figure 1).

  1. Table 1 needs an explanation of the identity of Items 1-16, especially for those readers not familiar with this methodology.

We have included the quote that was in the methodology in point 2.4 so that the reader can more easily access the meaning of each item. The items are as follows (if you take into account that we put them at the end of the table)

  1. Was the purpose stated clearly?
  2. Was relevant background literature reviewed?
  3. Was the sample described in detail?
  4. Was sample size justified?
  5. Were the outcome measures reliable?
  6. Were the outcome measures valid?
  7. Intervention was described in detail?
  8. Contamination was avoided?
  9. Co-intervention was avoided?
  10. Results were reported in terms of statistical significance?
  11. Were the analysis method(s) appropriate?
  12. Clinical importance was reported?
  13. Drop-outs were reported?
  14. The discussion should include how the results may influence clinical practice - do they offer

useful and relevant information about a client population, or an outcome of interest?

  1. Do they warrant further study?
  2. Conclusions were appropriate given study meth-ods and results

7: Table 1. Quality assessment of the studies included in the systematic review by McMaster critical review form – quantitative studies [36].

  1. Section 3.3 ‘This systematic review included ten experimental studies’ needs to be corrected to 17. The statement ‘All eight studies remained neutral benefits with a non-significant marginal benefit in a subset of patients after the intervention’ is a meaningless and misleading statement and needs to be removed. In the remaining 8 studies there was no evidence of benefit.

8: In the remaining eight studies there was no evidence of benefit …

  1. The structure of text in the Final paragraph is very confused which limits understanding of the meaning, this requires revision.

9: We have removed the confusing and poorly worded paragraph.

  1. Table 2. The total number of participants and number of participants per arm have a number of inconsistencies within the table and with respect to some of the publications. All of the data requires confirmation and revision where appropriate. In addition, ‘Duration’ needs to be changed to ‘Duration of Study’ and where appropriate (eg crossover studies) is different from Duration of Treatment.

10: We have reviewed each study in its entirety and corrected any errors that existed, which required a thorough review and for which we are grateful for your appreciation (Table 2). We have asterisked those that are crossover studies. And we clarify their duration at the foot of the table with an asterisk and the first author's initial).

  1. Since many aspects of lifestyle, eg diet, exercise, can influence antioxidant status of a person biology, it would be beneficial to the reader to be informed of the inclusion and exclusion criteria applied to recruitment of participants in all the studies.

11: We have reviewed the inclusion and exclusion criteria and they do not indicate such criteria, although there are several that have chronic alcohol or drug abuse and taking other types of antioxidants as exclusion criteria.

  1. In table 3 change increasement and decreasement with increase and decrease.

12: “Decrease in VAS and increase…”

  1. In the discussion of Coenzyme Q10, the last paragraph should provide some consideration of the observation suggesting in the studies of short duration pain and FIQ scores were significantly improved but the study of longer duration only pain was reduced. This could be consistent with beneficial effects waning rather than being maintained with continued use of Coenzyme Q10.

13: Overall, Q10 supplementation in the three studies [38, 42, 46] included in this review did not differ greatly in terms of the amount of CoQ10 300-400 mg/day administered. However, in terms of administration time, one of them [42] doubled the administration time compared to the other two. In addition to the need for a larger number of participants, we also highlight the need for a longer CoQ10 administration time for comparison.

  1. In the conclusion, it needs to be emphasized that significant improvement was only observed in a small majority, ie 53%, of studies so that the reader is not mislead with respect of the level of effectiveness demonstrated of antioxidant supplementation. The suggested ‘minimum time of a month and a half’, how was this period derived or calculated?

14: This review encompassed the literature and showed that the role of antioxidant supplements in FM could improve pain perception in patients, as measured by scales for this purpose.

Nine studies showed a significant improvement in pain perceived by FM patients that did the VAS scale and/or the FIQ scale. Specifically, …….

Furthermore, the number of weeks of supplementation appears to be positively associated with improved quality of life in subjects with FM, suggesting that a minimum time of 6 weeks of supplementation could have a functional impact on the efficacy of antioxidants on pain.

We have carried out an English revision of the entire text, which has been done by Martin Roche, a native English speaker.
